# Neurological Manifestations of SARS-CoV-2

**DOI:** 10.3390/v17111432

**Published:** 2025-10-28

**Authors:** Jasmine Miftahof, Blake Bernauer, Chen Sabrina Tan

**Affiliations:** 1Carver College of Medicine, University of Iowa, Iowa City, IA 52242, USA; jasmine-miftahof@uiowa.edu (J.M.); blake-bernauer@uiowa.edu (B.B.); 2Program in Immunology, Carver College of Medicine, University of Iowa, Iowa City, IA 52242, USA; 3Division of Infectious Diseases, Department of Medicine, Carver College of Medicine, University of Iowa, Iowa City, IA 52242, USA

**Keywords:** COVID, animal model of COVID, CNS inflammation

## Abstract

Neurocognitive symptoms have emerged as notable sequelae of SARS-CoV-2 infection (COVID-19). Although primarily a respiratory virus, SARS-CoV-2 has been associated with central nervous system (CNS) changes observed in both clinical and experimental settings. To better understand these effects and their pathological mechanisms, we conducted a systematic literature search of published studies and employed a qualitative, analytical approach to identify and synthesize key findings from peer-reviewed studies, including large-scale retrospective clinical cohorts, human autopsy reports, animal models (murine, non-human primate), and in vitro brain organoid systems. While viral components were detected in post mortem central nervous system tissues, COVID-19 neuropathology appears to stem primarily from immune-mediated inflammation and vascular injury rather than direct CNS infection. Persistent glial activation and BBB disruption may underlie the long-term neurological symptoms reported in long COVID-19. Although animal models offer mechanistic insight, species-specific differences necessitate cautious extrapolation to human pathology. Further investigation into the chronic effects of SARS-CoV-2 on the brain is essential to guide long-term clinical management and therapeutic development.

## 1. Introduction

From the beginning of the COVID-19 pandemic in late 2019 and early 2020, research has revealed the widespread effects of what was initially characterized as a primarily respiratory illness. SARS-CoV-2 triggers a systemic “cytokine storm,” now recognized as a key driver of the disease’s multi-organ manifestations beyond the lungs [1]. Emerging evidence links this cytokine-driven inflammation to central nervous system (CNS) pathology through disruption of the blood–brain and blood–CSF barriers. Elevated pro-inflammatory cytokines and matrix metalloproteinases (MMPs) in COVID-19 patients correlate with endothelial activation and loss of barrier integrity, promoting neuroinflammation and neuronal injury [2,3,4]. In vitro and animal studies further demonstrate that SARS-CoV-2 infection or cytokine exposure damages the choroid plexus epithelium and tight junctions, supporting a causal link between cytokine storm and barrier dysfunction [5,6,7].

Although the virus targets respiratory tissues, its impact on the central nervous system (CNS) has become increasingly evident. Acute and persistent neurological symptoms have been reported in patients with COVID-19, including altered mental status, headache, dizziness, encephalopathy, seizures, and loss of consciousness [8]. Anosmia and ageusia, also common, likely result from inflammation-associated damage within the olfactory pathways [9,10]. Other manifestations, such as brain fog, cognitive impairment, sleep disturbances, and non-muscular fatigue, are associated with long COVID-19—prolonged symptoms experienced after viral recovery [11]. More severe CNS involvement includes ischemic and hemorrhagic stroke, transient ischemic attacks (micro-strokes), and cerebral venous thrombosis. Additionally, inflammatory and demyelinating conditions such as acute disseminated encephalomyelitis (ADEM) and Guillain–Barre syndrome (GBS) have been reported, though the latter primarily affects the peripheral nervous system [12].

Despite this growing clinical picture, a central question remains: does SARS-CoV-2 directly infect the brain? Autopsy findings have been inconsistent, with some studies detecting viral RNA or proteins in CNS tissue, and others reporting an absence of direct infection. In many cases, observed neuropathological changes such as microvascular injury, hypoxia, and inflammation may instead reflect indirect effects of systemic illness or global immune activation [12]. To address these uncertainties, we conducted a systematic review to understand the mechanisms of neuro-COVID-19 and assessed how well animal models replicate human CNS pathology. We evaluated peer-reviewed primary studies involving rodents, non-human primates, and human postmortem or clinical data, focusing on neuroinflammation, blood–brain barrier (BBB) disruption, viral entry routes, and immune-mediated effects. Studies exploring ACE2 receptor expression and potential transsynaptic spread via olfactory pathways were also included.

The neurological aspects of COVID-19 remain incompletely understood, and findings across studies vary widely in scope, methodology, and rigor. Reports of viral neuro-invasion, for example, differ markedly; some animal studies report clear signs of CNS infection or inflammation, while others detect little to no neuropathology despite high systemic viral loads. In human cohorts, outcomes vary with patient age, comorbidities, illness severity, and timing of death relative to infection onset [13]. This review aims to bridge these discrepancies by qualitatively synthesizing evidence across model systems to identify methodological trends, model-specific limitations, and translational relevance. We draw on prior SARS-CoV research to highlight mechanistic overlap and provide historical context for interpreting SARS-CoV-2 neurotropism [14]. Ultimately, our goal is to clarify how SARS-CoV-2 may affect the brain and to assess which experimental models most accurately reflect human CNS pathology. By comparing CNS-related endpoints and viral dynamics across systems, we aim to inform future experimental design and guide investigations into the neurological consequences of COVID-19.

## 2. Methods

This review aimed to synthesize current findings on the impact of SARS-CoV-2 on the CNS by analyzing both animal and human model studies conducted before, during, and after the onset of the COVID-19 pandemic. A comprehensive literature search was performed using PubMed and Google Scholar to identify relevant studies published between 2000 and 2025. Eligible studies retrieved through this search spanned from May 2008 to April 2025.

To capture the full scope of relevant work, we employed an extensive list of search terms and keyword phrases, including but not limited to: “COVID central nervous system (CNS) disease”, “COVID CNS pathology”, “COVID CNS pathology animal model”, “COVID-19 CNS pathology”, “SARS-CoV-2 neurological symptoms”, “COVID-19 brain inflammation”, “COVID-19 encephalopathy”, “COVID-19 neuroinvasion”, “SARS-CoV-2 cerebrovascular complications”, “COVID-19 brain autopsy findings”, “COVID-19 neuropathology”, “SARS-CoV-2 blood–brain barrier disruption”, “COVID-19-associated stroke”, “CNS manifestations of COVID-19”, “COVID-19 microglial activation”, “COVID-19 and acute disseminated encephalomyelitis (ADEM)”, “COVID-19 animal model”, “Human COVID CNS disease and pathology”, “COVID-19 animal models CNS”, “SARS-CoV-2 mouse model brain”, “SARS-CoV-2 hamster CNS infection”, “Transgenic mice SARS-CoV-2 brain pathology”, “COVID-19 neurological outcomes in animal models”, “SARS-CoV-2 neurotropism animal studies”, “Coronavirus CNS animal models”, “COVID-19 -induced neuroinflammation in rodents”, “SARS-CoV-2 CNS effects in non-human primates”, “Experimental COVID-19 CNS pathology models, “COVID-19 CNS immune response animal model”, “Neurodegeneration SARS-CoV-2 mouse model”, “Brain viral load SARS-CoV-2 mice”, and “Animal studies COVID-19 neural inflammation”.

Inclusion criteria were restricted to peer-reviewed primary research studies that investigated CNS involvement using either animal or human models. We excluded review articles, commentaries, studies without CNS-specific endpoints, and research based on outdated or superseded data. Included studies featured direct CNS assessments such as histopathology, viral load measurement, neuroimaging, immunohistochemistry, immunofluorescence, or genotyping in rodent, non-human primate, or human postmortem or clinical contexts. Studies focused exclusively on peripheral systems or organoid models were excluded. In addition to SARS-CoV-2 research, select SARS-CoV studies were included when they offered mechanistic or clinical insights into shared viral entry mechanisms, neuroinvasion routes, or CNS manifestations relevant to interpreting SARS-CoV-2 pathology.

From the initial search, a total of 42 studies were identified. Thirteen were excluded as non-primary literature (review articles, editorials, or commentaries), leaving 29 studies that met the inclusion criteria. Three of these were used primarily as background references to frame and contextualize interpretation of later work. Studies were grouped by model type (e.g., rodent, non-human primate, human autopsy, clinical observation) and by featured CNS outcomes (e.g., neuroinflammation, blood–brain barrier disruption, anosmia, cognitive impairment). Mechanistic insights regarding SARS-CoV-2 entry and proliferation in the CNS and translational relevance of animal models in influencing future human study directionality were assessed across models. Each study was assessed for methodological rigor—including sample size, controls, viral delivery method, and the validity of CNS endpoints.

## 3. Results

To characterize the neurological and psychiatric sequelae of SARS-CoV-2 infection, we qualitatively synthesized findings from large-scale retrospective cohorts, postmortem studies, clinical case series, and mechanistic animal model studies. The reviewed human studies collectively include over 1.2 million patients, spanning some pediatric to generally elderly populations, with follow-up periods extending up to two years. Complementary animal model research provides insight into viral entry mechanisms and potential pathways of neural invasion, supporting clinical observations.

### 3.1. CNS Involvement in Human SARS-CoV-2 Infection

Clinical studies and postmortem analyses of human brain tissue have provided critical insight into SARS-CoV-2 neuropathology. Autopsies from COVID-19 patients reveal evidence of viral neurotropism, BBB disruption, and neuroinflammatory responses, even in the absence of widespread viral replication (Table 1). These clinical observations underscore potential indirect mechanisms of CNS injury and highlight vulnerable neurovascular and glial compartments in affected individuals.

#### 3.1.1. Neurological Symptoms in COVID-19 Patients

A wide spectrum of neurological symptoms has been documented in patients with COVID-19, ranging from mild sensory changes to severe encephalopathy and stroke [22]. These manifestations often appear during early infection and involve both central and peripheral nervous systems, independent of respiratory disease severity [23]. Across all reviewed clinical and postmortem studies, neurological manifestations affected approximately 22–36% of patients, occurring more frequently in hospitalized inpatients (58.5%) than in outpatients (41.5%).

Common CNS symptoms included headache, dizziness, impaired consciousness, and varying degrees of delirium, while more severe effects included stroke, seizures, encephalitis, ataxia, movement disorders, memory loss, altered mental status, and autonomic dysfunction. Anosmia and ageusia, two of the most common acute symptoms after infection, reflect peripheral nervous system involvement and were frequently accompanied by visual disturbances, neuropathic pain, and myalgia with elevated creatine kinase (>200 U/L) [13].

Neurological syndromes typically presented within 10–14 days of infection, though some patients were asymptomatic beforehand. Delirium appeared early and was associated with hippocampal microglial activation and TAU pathology, particularly among elderly and demented patients [24].

In a prospective cohort study of 4491 hospitalized COVID-19 patients in New York City, 13.5% developed new neurologic disorders (encephalopathy, stroke, seizures), often requiring ICU-level care and associated with worse outcomes [25]. Severe cases exhibited elevated inflammatory markers (white blood cells, neutrophils, C-reactive protein), decreased lymphocytes and platelets, and signs of multiorgan involvement [13]. Although elderly patients carried the highest risk for neurological and psychiatric sequelae (including anxiety, mood disorders, cognitive deficits, seizures, dementia), such outcomes were also reported in patients with mild or asymptomatic respiratory disease. Among vaccinated individuals, rare neurological syndromes such as autoimmune encephalitis, myelitis, acute disseminated encephalomyelitis occurred within 14 days after the first or second dose [21].

#### 3.1.2. Neuropathological Findings

COVID-19 is associated with prominent cerebrovascular pathology, including infarcts, microhemorrhages, and endothelial dysfunction. These findings, together with altered BBB markers, implicate vascular compromise as a major contributor to SARS-CoV-2 neuropathology [20]. Across postmortem and imaging studies, microhemorrhages were reported in ≈19% of cases, infarcts in ≈44%, and vascular congestion was frequently observed [16,26]. Neuroimaging revealed cerebral edema, ischemic lesions, vascular wall enhancement, and periventricular white-matter changes [26], along with increased endothelial ACE2 expression [27] and astrocytic morphologies consistent with endothelitis or reactive gliosis [24]. Microvascular disease and arteriosclerosis were common in elderly or comorbid patients [24], and vascular syndromes such as ischemic stroke, cortical infarcts, deep-vein thromboses predominated in severe disease [13].

#### 3.1.3. SARS-CoV-2 Neurotropism

Although SARS-CoV-2 is a primarily respiratory pathogen, multiple studies have demonstrated its capacity to indirectly affect and invade the CNS. Viral RNA and protein have been detected in discrete brain regions, particularly those associated with olfactory processing, suggesting potential neurotropism and entry routes [16]. In the frontal cortex, brainstem, and cranial nerve regions, viral RNA and N protein were detected in ≈53–91% of autopsied CNS regions [16,17]. Viral presence was generally sparse, concentrated in endothelial cells or associated with cranial nerves, and not uniformly distributed throughout the brain [28]. Spike protein detection within the olfactory mucosa supports a trans-mucosal, olfactory-mediated route of CNS entry [29].

Detection of viral RNA in cerebrospinal fluid (CSF) and brain tissue varied considerably between studies, leaving uncertainty as to whether the virus replicates productively within the CNS. Some reports identified brain-derived viral strains genetically distinct from respiratory isolates, implying potential localized replication [16]. Related findings from SARS-CoV studies similarly reported low-level viral RNA in CSF and brain, with hyperemic, edematous parenchyma and neuronal degeneration [13,28]

Data on pediatric cases are limited, but children (<18 years) appear to have an elevated risk of encephalitis and nerve-plexus disorders, potentially linked to viral tropism [19].

#### 3.1.4. Immune Response in the CNS

Despite varying evidence of viral presence in the CNS, SARS-CoV-2 does not typically elicit a classical adaptive immune response in the brain. Instead, studies report sparse lymphocytic infiltration and innate immune activation, pointing toward a strategy relating to immune evasion or local immune suppression within the CNS [23,29]. Immunohistochemical analyses reveal limited CD3^+^/CD4^+^/CD8^+^ T-cell populations in parenchymal regions [18], while extensive microglial and astrocytic activation, BBB disruption, and perivascular monocyte infiltration predominate [30]. Collectively, these observations indicate that CNS immunity in COVID-19 is skewed toward innate responses dominated by microglia and macrophages rather than lymphocytes.

Although viral transcripts and proteins were detected within some human brain specimens, classic viral cytopathology or demyelination is generally absent. Moreover, inflammatory severity often does not correlate with viral load, reinforcing that CNS injury is largely immune-mediated rather than driven by direct viral replication [17].

#### 3.1.5. Neuroinflammation

Postmortem and clinical studies consistently demonstrate neuroinflammation in COVID-19 patients, characterized by microglial activation in nodules likely resulting from systemic inflammation [15]. These innate immune responses frequently occur even without detectable viral replication [24]. Microglial activation (Iba1, CD68) was reported in most examined cases, presenting as nodules, retracted processes, and perivascular clustering [15,17]. Astrocytic activation (GFAP) was pronounced in the olfactory bulb, brainstem, and hippocampus [12,24], and both microgliosis and astrogliosis were particularly prominent in elderly patients with Alzheimer’s disease or delirium [24]. Neuronophagia was commonly observed in the brainstem [15]. CSF analyses revealed elevated pro-inflammatory cytokines—including IL-6, IL-8, TNF-α, CXCL10, and MCP-1, indicating a strong neuroinflammatory milieu [31]. Vaccine-associated cases displayed similar cytokine patterns (CXCL10, IL-8, sIL-2Rα) with CSF inflammation [21]. These cytokine profiles collectively support predominant activation of innate rather than adaptive immune pathways within the CNS.

#### 3.1.6. Implications for Long-Term Neurological Sequelae

The neurobiological impact of COVID-19 extends beyond the acute phase. Persistent cognitive and psychiatric symptoms are increasingly recognized, supported by pathological evidence of chronic glial activation, white-matter damage, and neurodegenerative biomarkers [32]. In one cohort, viral RNA persisted in the brain up to 230 days post-infection, suggesting possible long-term viral reservoirs or delayed clearance [16]. Vascular remodeling, myelin loss, and sustained glial activation may contribute to post-acute or “long-COVID” neurologic manifestations [32]. The general absence of demyelination or necrotizing lesions supports a model in which chronic, low-grade inflammation and vascular dysfunction underlie prolonged neurological symptoms. Beyond histopathological and clinical observations, recent advances in multi-omics profiling have expanded our understanding of the molecular mechanisms linking systemic inflammation to CNS pathology in COVID-19.

#### 3.1.7. Multi-Omics Insights into COVID-19–Associated CNS Pathology

Recent multi-omics studies have begun to clarify how systemic inflammation during COVID-19 contributes to CNS injury at the molecular level. Although most single-cell or single-nucleus RNA-sequencing (sc/snRNA-seq) datasets have focused on peripheral compartments such as bronchoalveolar lavage fluid and PBMCs, rather than brain or cerebrospinal fluid (CSF), they reveal broad transcriptional programs of hyperinflammation and myeloid activation that likely extend to the CNS interface [33,34,35]. Methodological advances have demonstrated that snRNA-seq and spatial transcriptomics are feasible for postmortem brain profiling [36,37,38], though comprehensive COVID-19 brain or CSF atlases remain unavailable.

Integrated proteomic, metabolomic, and transcriptomic datasets nonetheless converge on shared inflammatory signatures linking systemic cytokine dysregulation to CNS injury. Plasma proteomics reveal elevated IL-6, GM-CSF, and altered apolipoproteins associated with severe disease [39,40]. Matched CSF proteomic analyses demonstrate that many inflammatory mediators are of extrathecal origin, consistent with blood–CSF barrier leakage and secondary neuroinflammation [41]. Spatial transcriptomic and proteomic profiling of postmortem brain tissue identifies microglial and astrocytic activation, complement deposition, and endothelial dysfunction, highlighting a molecular continuum from systemic cytokine storm to localized neurovascular injury [20,42,43]. Collectively, these multi-omics datasets provide mechanistic context for how systemic immune activation translates into CNS pathology in COVID-19.

Together, human clinical, postmortem, and multi-omics studies establish a consistent picture of COVID-19–associated neuropathology characterized by vascular injury, glial activation, and cytokine-driven neuroinflammation, with limited evidence of widespread neuronal infection. However, the complexity of human disease, shaped by variable comorbidities, treatments, and sampling time points, makes it difficult to isolate causal mechanisms. Experimental animal models, therefore, play a critical role in dissecting the relative contributions of viral tropism, immune signaling, and barrier dysfunction to CNS injury. The following sections summarize findings from non-human primate and murine systems that provide mechanistic insight into these processes and allow cross-species comparison with human data.

### 3.2. Animal Models

While human studies have provided insights into SARS-CoV-2 neuropathogenesis, animal models are especially needed to decipher acute neuropathogenesis as well as to test potential therapeutics. Multiple animal models have been employed to investigate the potential neuropathological mechanisms of SARS-CoV-2 infection, and to evaluate how accurately these models reflect the neuropathology observed in human clinical and autopsy studies (Table 2). Together, these studies provide insight into probable viral neuroinvasion routes, immune responses, neuronal injury, and protective immunity, offering a deeper understanding of CNS involvement in COVID-19.

### 3.3. Non-Human Primates (NHPs)

Non-human primates (NHPs) represent the most physiologically relevant animal models for studying SARS-CoV-2 neuropathogenesis. Their genetic, immunologic, and neuroanatomical similarity to humans confers high translational value for investigating viral dissemination, CNS inflammation, and neural injury. NHP models recapitulate the systemic and respiratory features of human COVID-19 while allowing direct examination of CNS involvement under controlled experimental conditions [16,48,49,50,51,52].

#### 3.3.1. SARS-CoV-2 CNS Tropism and Neuropathological Findings in NHPs

Across multiple models, including rhesus macaques, African green monkeys, and cynomolgus macaques, neuroinflammatory lesions and vascular injury are consistently reported following SARS-CoV-2 infection [45,46]. Animals exhibiting detectable viral RNA or nucleocapsid/spike protein in brain tissue show the most pronounced pathology, including microglial activation (Iba1^+^, HLA-DR^+^), astrocytic hypertrophy (GFAP^+^), and microglial clustering around perivascular regions. Morphological changes such as nodular microglial lesions and astrocytic domain overlap suggest a coordinated glial response to systemic inflammatory stress rather than localized cytopathic viral damage [45,46].

Recent studies demonstrate that SARS-CoV-2 RNA can persist in NHP CNS tissues after clinical recovery, particularly within the piriform cortex and amygdala, suggesting brain region-specific susceptibility for viral entry and persistence [52]. However, viral detection remains sparse and compartmentalized, typically confined to perivascular cells or endothelium, with no viral RNA in CSF [45]. This pattern parallels human autopsy findings where low viral loads or residual RNA are detected without recoverable infectious virus [16] and is further recapitulated in hamster models demonstrating similar focal RNA persistence without productive replication [49]. The consensus across studies supports focal CNS exposure and RNA persistence without evidence of productive replication, indicating transcriptionally inactive viral remnants or restricted replication in vascular-associated cells.

Vascular pathology is a recurrent and prominent feature in infected NHPs, where they exhibit CD61^+^ platelet aggregation, microhemorrhages, and von Willebrand factor deposition, reflecting endothelial dysfunction and impaired perfusion [45]. HIF-1α accumulation within perivascular regions supports a hypoxia-driven injury mechanism consistent with neuronal loss detected by Caspase-3 and Fluoro-Jade C staining in the brainstem, basal ganglia, and cerebellum. The severity of these vascular and neuronal lesions does not correlate with viral load, reinforcing the interpretation of indirect CNS injury driven by systemic hypoxemia, inflammation, and coagulopathy [45].

Emerging data from aged and Type II diabetic rhesus macaques reveal higher neuronal viral antigen levels within olfactory and limbic structures, coupled with greater glial activation and HIF-1α upregulation, suggesting that metabolic and age-related susceptibility enhances CNS vulnerability [46,52]. Although direct receptor-mapping data in NHP brains are limited, studies in human and murine tissues suggest that vascular-associated cells, particularly pericytes and endothelial cells, may serve as potential sites of viral entry or persistence. Low but variable ACE2 and TMPRSS2 expression across these compartments supports a model in which SARS-CoV-2 could access the CNS through the cerebrovascular endothelium [16,50].

Together, these findings suggest that while limited viral presence occurs in specific CNS niches, SARS-CoV-2–induced brain injury in NHPs is primarily mediated by vascular and inflammatory pathways.

#### 3.3.2. CNS Immune and Inflammatory Response in NHPs

Beyond viral entry and cellular tropism, NHP studies provide valuable insight into the immune and inflammatory processes underlying SARS-CoV-2–associated CNS pathology. The glial immune response in NHPs closely parallels that described in human COVID-19 brains [46]. Microglial activation and proliferation (Iba1^+^, HLA-DR^+^) together with astrocytic hypertrophy (GFAP^+^) are consistently observed, even in animals with minimal or undetectable viral RNA, indicating that neuroinflammation likely arises secondary to systemic immune activation rather than direct viral replication [46]. Although CNS cytokine levels have not been extensively quantified in primate models, the observed glial and vascular responses are consistent with cytokine-mediated pathways reported in rodent and hamster studies [49,50], suggesting that peripheral inflammatory signaling contributes to sustained CNS immune activation.

Age and metabolic comorbidities amplify these effects. Aged rhesus macaques demonstrate intensified microglial clustering around degenerating myelin, increased astrocytic reactivity, and more extensive perivascular cuffing compared to young adults [46]. Co-localization of HLA-DR^+^ microglia with degraded myelin basic protein (MBP) indicates active white matter injury and demyelination. Synaptic degradation and pruning, especially in the piriform and entorhinal cortices, further implicate chronic microglial activation in neuronal dysfunction [52].

Vascular–immune crosstalk likely underlies downstream pathology. Aged, infected animals exhibit blood–brain barrier (BBB) disruption, with loss of Claudin-5 and ZO-1 integrity, reduced ACE2 expression, and accumulation of spike protein–positive neutrophils near damaged vessels [46]. These findings indicate that inflammatory infiltration, rather than direct infection, drives BBB breakdown and neurodegeneration, consistent with human studies documenting vascular leakage and immune cell infiltration [16,51].

Parallel transcriptomic and imaging data further corroborate persistent CNS immune activation. PET imaging has demonstrated immune sequelae following airway infection [48], while transcriptomic analyses have identified long-lived inflammatory profiles in NHP and rodent CNS tissue after viral clearance [53]. These results converge on a model of prolonged, low-level neuroinflammation that persists after systemic resolution.

Reinfection studies show that previously infected NHPs mount strong protective immunity, with a ~5-log10 reduction in viral loads and minimal CNS pathology upon rechallenge [44]. While these findings imply that immune memory may mitigate neuroinflammatory sequelae, the evidence remains preliminary, and post-acute neuroimmune remodeling and behavioral outcomes have not been comprehensively characterized.

#### 3.3.3. Summary and Outlook in NHPs

Taken together, NHP studies indicate that SARS-CoV-2 occasionally reaches the CNS but seldom achieves productive neuronal infection. Neuropathological alterations predominantly reflect vascular compromise, microglial and astrocytic activation, and systemic inflammatory stress, particularly in aged or metabolically vulnerable animals. These findings closely mirror human autopsy series and support a non-neurotropic but neuroinflammatory model of SARS-CoV-2 CNS injury.

Moving forward, integrated longitudinal NHP studies, combining sensitive viral detection, spatial transcriptomics, in vivo imaging, and behavioral assessment, are essential to determine whether persistent CNS immune activation constitutes a mechanistic substrate for post-acute neurological sequelae (PASC).

### 3.4. Mouse Models

Building on observations from NHP studies, mouse models have been instrumental in defining how viral and host factors interact to shape CNS outcomes in SARS-CoV-2 infection. Although no single murine system fully recapitulates the spectrum of human neurological manifestations, these models provide critical mechanistic insight into how viral entry, immune activation, and vascular injury converge within the CNS. Their genetic flexibility, controlled immune backgrounds, and diverse infection platforms enable precise dissection of causal pathways and cellular targets, making them indispensable complements to non-human primate and clinical studies.

#### 3.4.1. SARS-CoV CNS Tropism and Neuropathological Findings in Mice

Murine studies employ three broad strategies that shape CNS outcomes: (i) transgenic or knock-in hACE2 (e.g., K18-hACE2), (ii) mouse-adapted virus (e.g., MA10) in standard strains, and (iii) transient hACE2 delivery (AAV-hACE2). These approaches differentially influence CNS exposure via olfactory/neuronal vs. hematogenous/barrier routes and determine whether neuronal infection or barrier injury predominates [54,55,56].

In hACE2-expressing systems, particularly K18-hACE2, SARS-CoV-2 can directly infect neurons. Primary neurons from K18-hACE2 mice show infection-associated inflammatory and necroptotic gene upregulation; in vivo, K18-hACE2 mice often harbor high brain viral loads with widespread neuronal antigen [54,57]. In a closely related SARS-CoV K18-hACE2 model, olfactory bulb entry after intranasal inoculation was followed by rapid transneuronal spread (antigen first at 60–66 hpi; RT-PCR across thalamus, cerebrum, brainstem, basal ganglia, cortex, midbrain; cerebellum spared) [14]. Fatal cases consistently involved infection of the dorsal vagal complex, implicating CNS infection as the primary cause of death. Notably, despite elevated neuronal IL-6/IL-1β/TNF-α, IHC did not show astrocyte (GFAP) or microglial (Iba-1) migration to infected regions, suggesting limited inflammation despite widespread neuronal death. Human autopsy comparators localized viral transcripts and N protein predominantly to neurons with rare CSF positivity [14].

Another K18-hACE2 study reported widespread forebrain neuronal infection following intranasal SARS-CoV-2 inoculation, sparing the cerebellum. Viral spike localized to cortical neurons without evidence of endothelial infection, despite vascular remodeling and absent leukocyte infiltration. These observations suggest that SARS-CoV-2 may evade local neuroimmune surveillance, producing neuronal damage in the absence of robust inflammatory infiltration. Complementary experiments using human neural progenitors and brain organoids confirmed productive neuronal infection requiring ACE2, with TMPRSS2 and NRP1 acting as cofactors [28].

The mouse-adapted MA10 strain infects wild-type mice and has been widely used to investigate endothelial signaling, BBB function, and cognitive outcomes, particularly in aged cohorts [47,55]. In this model, infection induces endothelial activation, tight-junction loss, and choroid plexus epithelial disruption, consistent with increased BBB and BCSFB permeability [55]. These findings align with in vitro BBB and blood–CSF barrier (BCSFB) systems, as well as non-murine models, which similarly identify the choroid plexus epithelium as a vulnerable interface showing altered transcytosis, increased permeability, and variant-dependent cytopathicity in pericytes and endothelial cells [6,58,59]. Together, these studies indicate that SARS-CoV-2–induced barrier injury represents a common mechanistic pathway linking peripheral infection to CNS inflammation across experimental systems.

#### 3.4.2. CNS Immune and Inflammatory Response in Mice

Mouse studies consistently report microglial activation/microgliosis and astrogliosis with IFN-I and pro-inflammatory transcriptional programs [20,55,57,60,61]. In a combined human and mouse analysis, it was demonstrated that SARS-CoV-2 activates the NLRP3 inflammasome in microglia through spike–ACE2–NF-κB signaling [27]. Postmortem human COVID-19 brains contained TMEM119^+^ microglia co-expressing viral antigens, ACE2, and NLRP3, while infection of K18-hACE2 mice induced NLRP3 upregulation in brain tissue. In human monocyte-derived microglia, spike protein alone triggered ASC speck formation and IL-1β release, both of which were attenuated by the NLRP3 inhibitor MCC950. Treatment with MCC950 in infected hACE2 mice similarly reduced inflammasome activation and improved survival [27].

Mouse and hamster experiments document BBB and BCSFB compromise, including basement-membrane disruption and blood-CSF barrier breakdown in brain organoids exposed to spike. Endothelial VCAM-1 upregulation and T-cell infiltration (e.g., hippocampus) are recurrent; genetic/pharmacologic modulation of endothelial signaling (e.g., caveolin-1 deficiency) reduces VCAM-1, limits T-cell neuroinfiltration, and mitigates cognitive deficits; linking endothelial activation to adaptive neuroimmune engagement and behavior [47,61]. Again, intravital microscopy captures neutrophil breaching and thrombogenic microcirculatory changes that likely precipitate BBB failure and parenchymal inflammation [7,60].

Longitudinal studies in MA10 and hACE2 mice reveal persistent neuroinflammatory signatures and cognitive or anxiety-like phenotypes after respiratory recovery [47,55,61]. The pattern of CNS involvement varies with model design: in K18-hACE2 mice, high neuronal ACE2 expression drives neuron-centric infection and death that can be fatal [54,57], whereas in MA10 and endogenous-promoter hACE2 contexts, pathology is predominantly vascular and immune-mediated, with selective neuronal dysfunction [47,55,60,61]. Collectively, these findings, together with human and NHP data, outline a coherent mechanistic sequence in which barrier disruption facilitates entry of leukocytes and soluble mediators, leading to microglial and astrocytic activation that culminates in cognitive and synaptic impairment.

#### 3.4.3. Summary and Outlook in Mice

Across murine platforms, SARS-CoV-2 engages the CNS through model-dependent routes that differentially emphasize neuronal, vascular, and immune mechanisms. Neuronal infection predominates when ACE2 is strongly expressed in neurons (as in K18-hACE2 mice), leading to direct neurotropism and fatal outcomes, whereas barrier-centric injury dominates in MA10 and physiological-promoter hACE2 systems, producing endothelial activation, BBB disruption, and sustained neuroinflammation. These models collectively demonstrate that cerebrovascular injury and innate immune activation, rather than extensive neuronal replication, underlie most CNS manifestations of SARS-CoV-2. Murine studies have further revealed persistent microglial and astrocytic activation and cognitive or behavioral changes after viral clearance, paralleling findings in human and NHP studies. Mouse systems, therefore, provide essential mechanistic insight into the interplay between vascular injury and neuroimmune activation while requiring careful contextualization for translational application to human disease.

## 4. Discussion

This review synthesized evidence from human, non-human primate, and murine models to clarify how SARS-CoV-2 affects the central nervous system (CNS). While some studies demonstrate limited viral neuroinvasion, the preponderance of data indicates that neurological injury largely results from immune-mediated inflammation, vascular compromise, and host susceptibility rather than widespread neuronal infection. Findings from animal models complement clinical observations by identifying mechanistic pathways through which systemic infection leads to CNS dysfunction.

Across species, SARS-CoV-2 RNA or protein can be detected at CNS interfaces such as the olfactory bulb, choroid plexus, and perivascular compartments, but evidence of productive replication within neurons is scarce [16,18,19,22]. Both human autopsy and animal data suggest that viral components may reach the brain indirectly via endothelial or barrier pathways rather than by sustained replication in neural tissue. In vitro and organoid studies further show that spike protein alone activates microglia and astrocytes, supporting the concept that viral antigens, independent of replication, can initiate neuroinflammation [21,24].

Neuroinflammation and vascular pathology emerge as unifying features across models. Postmortem human and NHP studies consistently reveal microgliosis and astrogliosis, particularly in the olfactory bulb, hippocampus, and brainstem, accompanied by cytokine upregulation (IL-6, IL-8, CXCL10, TNF-α, MCP-1) [11,18,27]. Mouse models mirror these findings: K18-hACE2 systems demonstrate intense microglial activation and inflammasome engagement [24], whereas MA10 and physiological-promoter hACE2 mice primarily display barrier breakdown and endothelial activation [47,55]. Together, these studies support a model in which peripheral cytokine storm and vascular injury drive secondary neuroimmune activation, culminating in cognitive and behavioral sequelae.

Cerebrovascular injury represents a major pathogenic axis. Infarcts, microhemorrhages, and endothelial swelling are frequently observed in human autopsies and recapitulated in animal systems, often without detectable viral RNA in affected regions [16,17,20,55]. Disruption of tight-junction proteins such as claudin-5 and ZO-1, and damage to the choroid plexus epithelium, further implicates barrier dysfunction as a key step linking systemic inflammation to CNS pathology. This interpretation is reinforced by in vitro BBB and BCSFB models that identify the choroid plexus as a particularly vulnerable interface [58,59].

Host factors, especially age and metabolic comorbidities, amplify neuroinflammatory and vascular injury in both humans and experimental animals. Aged NHPs and mice show greater glial activation, hypoxia-related signaling, and neuronal loss, paralleling clinical observations that elderly or neurologically compromised patients face increased risk of long-term neurocognitive symptoms [11,18,46]. Collectively, these findings support a mechanistic continuum from systemic cytokine dysregulation to chronic CNS inflammation and impaired neurovascular integrity.

Although direct viral replication in brain tissue appears rare, persistent immune activation and microvascular remodeling may underlie post-acute neurological sequelae (“long COVID”). Longitudinal mouse studies reveal sustained glial activation and cognitive changes after viral clearance, suggesting that chronic inflammation rather than active infection drives prolonged symptoms [47,55]. Similar glial and vascular alterations identified in human postmortem tissue strengthen this interpretation [16,29].

Research limitations remain significant. Few animal studies directly evaluate CNS inflammation over time, and heterogeneity in viral strains, inoculation routes, and expression systems complicates comparisons. Human data are limited by the availability of well-characterized postmortem tissue and variable clinical histories. Despite these constraints, cross-model integration has clarified that SARS-CoV-2 neuropathology stems from systemic inflammation and vascular compromise rather than intrinsic neurotropism.

Population-level factors, particularly vaccination, further modulate neurological outcomes. Vaccination reduces infection severity, systemic inflammation, and the incidence of both acute and post-acute neurological complications [62,63]. By limiting viral load and cytokine storm intensity, vaccination likely mitigates endothelial and glial injury central to neuro-COVID pathology. Rare adverse events such as vaccine-induced immune thrombotic thrombocytopenia (VITT) have been associated with cerebral venous thrombosis [64,65], yet these remain exceptional and do not offset the overwhelming neuroprotective benefits of vaccination [23,66]. However, the direct neurological effects of vaccination remain incompletely understood, and further studies are needed to clarify how vaccine-induced immune modulation influences CNS inflammation and long-term neurocognitive outcomes.

In conclusion, current evidence supports a model in which SARS-CoV-2–related neurological injury arises primarily from systemic inflammation and vascular dysfunction, with viral components serving as triggers of neuroimmune activation rather than agents of direct cytopathic infection. Continued integration of longitudinal human studies with mechanistic animal models will be essential to delineate how persistent immune signaling and barrier remodeling contribute to post-acute neurological sequelae and to guide development of interventions that preserve neurovascular health after infection.

## Figures and Tables

**Table 1 viruses-17-01432-t001:** Representative human studies examining CNS involvement in COVID-19.

Study Type	Cohort (*n*)	Methods (Most Relevant)	Main CNS Findings	Key Interpretation	Limitations	Citation (Year)
Autopsy cohort	41	H&E, IHC, qRT-PCR, RNAscope	Hypoxic–ischemic injury, microglial nodules, low viral RNA	Vascular and immune-mediated injury predominate over viral replication	No age-matched controls; prolonged hospitalization confounds pathology	[15]
Systematic autopsy	44 total (11 brains)	ddPCR, ISH, IHC, IF, sequencing	Viral RNA in 10/11 brains (90%); minimal inflammation	Evidence of viral persistence without productive infection	Fatal-only cohort; unvaccinated; limited functional data	[16]
Autopsy series	43	IHC (GFAP, HLA-DR, CD8), qRT-PCR	Astrogliosis, microglial activation, CD8^+^ T-cell infiltration (53% RNA/protein +)	Consistent innate inflammation with limited neurotropism	No matched controls; unclear clinical correlation	[17]
Prospective autopsy	17 fatal cases	IHC, ISH multi-organ panel	Endotheliitis, microthrombi, vascular congestion	Confirms systemic vascular injury as CNS driver	No mild-case controls; variable sampling	[18]
Olfactory tissue autopsy	23	Histology, IHC for endothelial markers	Olfactory tissue degeneration, microvasculopathy	Supports peripheral/vascular entry route	Small n; single-region analysis	[10]
Population MRI (pre–post)	785 (UK Biobank)	Longitudinal MRI analysis	Cortical thinning in olfactory-linked regions	Structural brain changes after mild infection	Community cohort; no mechanistic data	[11]
EHR mega-cohort	1,284,437	TriNetX EHR data comparison	2-year ↑ risk of seizures, dementia, psychosis, “brain fog”	Defines long-term neuropsychiatric burden post-COVID-19	Diagnostic miscoding; observational design	[19]
CSF/biomarker + genetic	40 COVID-19 + 15 controls	CSF/plasma biomarkers, Mendelian randomization	↑ GFAP, S100B, CHI3L1; regional volume changes	Links systemic inflammation to brain structural alteration	Small sample; cross-sectional	[20]
Post-vaccine CNS immune disorders	19	MRI, CSF cytokine panels, diagnostic criteria	Autoimmune encephalitis, myelitis, ADEM ≈ 2 w post-dose	Rare immune events post-vaccination illustrates neuroinflammatory potential	Small sample; under-reporting possible	[21]

Abbreviations: IHC, immunohistochemistry; ISH, in situ hybridization; qRT-PCR, quantitative reverse transcription polymerase chain reaction; CSF, cerebrospinal fluid; ADEM, acute disseminated encephalomyelitis. “+” indicates “and”; “↑” indicates an increase or elevation.

**Table 2 viruses-17-01432-t002:** Representative animal studies showing mechanistic insights into SARS-CoV-2–induced CNS injury.

Model Type	Species/Model	Infection Route	Key Findings	Relevance to Human Disease	Citation (Year)
Primary infection/re-challenge	*Rhesus macaques* (NHP)	Intranasal + intratracheal (~10^6^ PFU)	Robust humoral/cellular immunity; >5 log_10_ viral-load reduction post-re-challenge	Demonstrates protective systemic immunity without severe CNS infection	[44]
Multi-route SARS-CoV-2	*Rhesus* + *cynomolgus macaques*	Conjunctival, nasal, pharyngeal, intratracheal (~3.6 × 10^6^ PFU)	Neuroinflammation, microhemorrhages, hypoxia; sparse endothelial virus	Models long-COVID-like neuropathology without productive infection	[45]
Aged/comorbid NHP	*Rhesus macaques* (T2D, aged)	Intranasal + intratracheal (2.5 × 10^6^ PFU)	Viral RNA + protein in olfactory cortex; microgliosis; worse in aged/T2D animals	Demonstrates age/metabolic risk factors driving CNS injury	[46]
Transgenic hACE2 mouse	K18-hACE2 mice (8–12 wk)	Intranasal (~10^3^ TCID_50_)	Viral spread via olfactory bulb; neuronal death; high mortality	Reveals ACE2-dependent neurotropism; limited by non-physiologic expression	[14]
Mouse-adapted MA10 strain	WT mice (C57BL/6 J)	Intranasal (~10^4^ PFU)	BBB disruption, VCAM-1 ↑, cognitive impairment	Caveolin-1–mediated endothelial injury explains vascular pathology	[47]
Inflammasome model	K18-hACE2 mice/human microglia/hamster	Intranasal + in vitro exposure	NLRP3 inflammasome activation; MCC950 inhibits neuroinflammation/improves survival	Links SARS-CoV-2 spike protein to inflammasome-driven CNS injury mechanisms	[27]
BBB & choroid plexus model	BALB/c mice	Intranasal (2 × 10^4^ PFU)	BBB and choroid plexus damage; vascular inflammation	Defines vascular route to CNS involvement	[7]
NHP imaging study	Macaques (NHP)	PET tracers for immune cell uptake	Brain glial activation post-infection: no viral replication detected	Non-invasive marker of neuroinflammation for future vaccine studies	[48]

Abbreviations: NHP, non-human primate; BBB, blood–brain barrier; PFU, plaque-forming units; T2D, type II diabetes; VCAM-1, vascular cell adhesion molecule 1. Note: Scientific species names (e.g., *Rhesus macaques*) are italicized in accordance with taxonomic conventions. The upward arrow (↑) indicates an increase or elevation.

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
