# Peer review of "Neurological Manifestations of SARS-CoV-2"

_viruses, 2025, doi:10.3390/v17111432_

Round 1
Reviewer 1 Report
Comments and Suggestions for Authors
This manuscript presents a systematic review on the impact of SARS-CoV-2 on the central nervous system (CNS). The authors performed a literature search and integrated data from retrospective clinical studies, human autopsy findings, animal models (mice and non-human primates), and in vitro brain organoid systems. The review primarily focuses on the potential routes of viral entry into the CNS, neuroinflammation, glial cell activation, blood–brain barrier (BBB) disruption, and the pathological mechanisms underlying long-term neurological symptoms.
The main conclusions include:
- Limited and inconsistent viral detection: Some autopsy studies reported RNA or protein signals, but no evidence has confirmed the replication of intact viral particles in brain tissue or cerebrospinal fluid.
- Neuroinflammation and vascular injury as key mechanisms: Widespread glial activation, elevated inflammatory cytokines, BBB disruption, and vascular pathology suggest that immune-mediated damage and vascular injury are the main drivers.
- Insights and limitations from animal models: Humanized mouse and NHP studies indicate potential invasion routes (e.g., olfactory pathway, ACE2 expression), but interspecies differences limit direct extrapolation.
- Host susceptibility: Advanced age and comorbidities substantially increase the risk of CNS involvement.
- Long-term sequelae (Long COVID): Persistent glial activation, sustained BBB disruption, and vascular remodeling may underlie cognitive deficits and psychiatric manifestations.
Overall, the review emphasizes immune- and vascular-mediated pathology, rather than direct neuroinvasion, as the core mechanism of COVID-19 neuropathology, and highlights the need for further studies on its long-term effects and potential therapeutic interventions.
Major Comments
- Abstract length: The abstract is overly long. It should be condensed to 250–300 words, with a clearer emphasis on the research question, methods, main findings, and conclusions.
- Literature update: Most cited references are from 2021–2022. More recent studies (2023–2025), especially on animal models and Long COVID, should be incorporated to ensure the review is up to date.
- Animal models section: The number of references is relatively limited, and some general conclusions are drawn from single experimental reports. A more balanced and systematic discussion with comparative insights is recommended.
- Terminology consistency: The term “COVID” should be standardized to “COVID-19” throughout the text to ensure formal academic usage.
- Introduction: The description of the “cytokine storm” requires additional references and further clarification of its link to CNS pathology.
- Vaccination effects: The potential impact of vaccination on COVID-19-related neuropathology has not been discussed. Considering its possible protective or exacerbating roles, this omission represents a significant gap.
- Multi-omics evidence: The review currently lacks discussion of multi-omics studies (e.g., single-cell sequencing, transcriptomics). Incorporating these data would strengthen the mechanistic understanding of CNS immunopathology.
Author Response
We agree with the reviewers' comments and suggestions and we have now extensively revised our manuscript. Please see the attached file for point specific responses.

Reviewer 2 Report
Comments and Suggestions for Authors
The manuscript presents a comprehensive and well-structured synthesis of current evidence regarding the neurocognitive sequelae of SARS-CoV-2 infection. By integrating data from diverse sources, including clinical cohorts, human autopsy studies, animal models, and in vitro brain organoid systems, it provides a balanced perspective on the mechanisms potentially underlying COVID-19-associated neurological manifestations. This is a scientifically rigorous and timely review that not only synthesizes current evidence but also sets clear priorities for advancing our understanding of SARS-CoV-2 neuropathology. It is suitable for publication and will be of interest to both clinicians and researchers working in neurovirology, neuropathology, and long COVID.
A key strength of the work is its critical evaluation of evidence for direct CNS viral invasion versus indirect, immune-mediated, and vascular mechanisms. The authors appropriately emphasize that while sporadic detection of viral RNA and proteins suggests possible neurotropism, the absence of intact virions in cerebrospinal fluid and brain tissue strongly argues against SARS-CoV-2 as a robust neuroinvasive virus. Instead, the converging evidence highlights neuroinflammation, glial activation, and blood-brain barrier disruption as primary drivers of neuropathology.
The review also highlights the significance of host factors, particularly age and comorbidities, in influencing CNS vulnerability. This adds translational relevance, linking experimental observations to clinical populations at the highest risk for long-term neurological complications. Importantly, the discussion of limitations in transgenic mouse models (e.g., hACE2 overexpression not mirroring human physiology) demonstrates appropriate caution in extrapolating preclinical findings. Overall, the manuscript makes a significant contribution to the field by clarifying that COVID-19 neuropathology is more consistent with immune-mediated and vascular injury than with direct viral infection of neural tissue. The focus on persistent glial activation and BBB disruption provides a plausible mechanistic framework for the long-term cognitive and neurological symptoms observed in long COVID.
Author Response
We agree and have now revised the manuscript.

Reviewer 3 Report
Comments and Suggestions for Authors
The impact of COVID-19 on the central nervous system has long been a concern as a post-infection symptom of COVID-19 infection. This review, compiled by the authors, effectively summarizes research using various animal models. I found this review extremely informative and grateful. I'm sure many readers will find this review equally useful. Importantly, the review demonstrates that SARS-CoV-2 does not directly infect the central nervous system, but rather causes vascular damage within the central nervous system through systemic inflammation and vascular dysfunction. Whereas this animal model cannot be immediately applied to human pathology, it will likely provide important insights for clinical application in humans.
As the authors have already noted, this review has limitations. Unfortunately, the number of animal experiments addressing inflammation in the central nervous system is limited compared to the abundance of available clinical and anatomical data. Even so, this review is certainly interesting and useful. I wish this limitation had been addressed more in the discussion section.
Again, this review is a really interesting and useful paper.
Author Response
We agree and have revised the manuscript.

Round 2
Reviewer 1 Report
Comments and Suggestions for Authors
I have no further questions and accept it for publication in present form